# RNA-Seq is not required to determine stable reference genes for qPCR normalization

**Nirmal Kumar Sampathkumar**[1,2☯¤a]\*, **Venkat Krishnan Sundaram**[3,4☯]\*, **Prakroothi S. Danthi**[5], **Rasha Barakat**[3,6], **Shiden Solomon**[1,2], **Mrityunjoy Mondal**[1,2¤b], **Ivo Carre**[1,2], **Tatiana El Jalkh**[3], **Aïda Padilla-Ferrer**[3], **Julien Grenier**[3], **Charbel Massaad**[3], **Jacqueline C. Mitchell**[2]

**1** UK-Dementia Research Institute, King's College London, London, United Kingdom, **2** Department of Basic and Clinical Neuroscience, Institute of Psychiatry, Psychology and Neuroscience, Basic and Clinical Neuroscience Institute, King's College London, London, United Kingdom, **3** INSERM U1124, Université de Paris, Faculty of Basic and Biomedical Sciences, Paris, France, **4** Paul Flechsig Institute of Neuropathology, University Clinic Leipzig, Leipzig, Germany, **5** McGill Group for Suicide Studies, The Douglas Research Center, McGill University, Montréal, Canada, **6** INSERM U1016, Institut Cochin, Paris, France

☯ These authors contributed equally to this work.
¤a Current address: Alzheimer's Research UK Oxford Drug Discovery Institute, Centre for Medicines Discovery, University of Oxford, Oxford, United Kingdom.
¤b Current address: Deutsches Zentrum für Neurodegenerative Erkrankungen, Bonn, Germany.
\* nirmal.sampathkumar@cmd.ox.ac.uk (NKS); venkatkrishnan.sundaram@u-paris.fr (VKS)

**Data Availability Statement:** The original contributions presented in the study are included in the article & Supplementary Data. The metadata and source data for the RNA-Seq and qPCR

## Abstract

Assessment of differential gene expression by qPCR is heavily influenced by the choice of reference genes. Although numerous statistical approaches have been proposed to determine the best reference genes, they can give rise to conflicting results depending on experimental conditions. Hence, recent studies propose the use of RNA-Seq to identify stable genes followed by the application of different statistical approaches to determine the best set of reference genes for qPCR data normalization. In this study, however, we demonstrate that the statistical approach to determine the best reference genes from commonly used conventional candidates is more important than the preselection of 'stable' candidates from RNA-Seq data. Using a qPCR data normalization workflow that we have previously established; we show that qPCR data normalization using conventional reference genes render the same results as stable reference genes selected from RNA-Seq data. We validated these observations in two distinct cross-sectional experimental conditions involving human iPSC derived microglial cells and mouse sciatic nerves. These results taken together show that given a robust statistical approach for reference gene selection, stable genes selected from RNA-Seq data do not offer any significant advantage over commonly used reference genes for normalizing qPCR assays.

## Author summary

RTqPCR is a powerful technique that is widely used to quantify gene expression in research and diagnostics of different diseases. The technique involves making multiple copies (amplification) of a specific target DNA. The amplified target DNA binds to a

experiments can be accessed at https://doi.org/10.6084/m9.figshare.15169104.v2. The datasets available include the normalized count matrices for the RNA-Seq data, qPCR Cq values and differential expression analysis, primer sequences, standard curves and amplification efficiencies, Coefficient of Variation and NormFinder analysis.

**Funding:** The author(s) received no specific funding for this work.

**Competing interests:** The authors have declared that no competing interests exist.

molecule that emits fluorescence upon binding. The extent of fluorescence correlates to the amount of DNA present. To precisely quantify this fluorescence (and thus the quantities of target DNA), internal control genes also called as reference genes need to be determined. Such genes, in principle, do not have varied expression across samples and would exhibit the same fluorescence in all samples. They can thus be used to normalize the expression of the Target DNA. Unfortunately, choosing the right reference gene is very tricky and poor choice of reference genes results in unreliable data both in research and in diagnostics. In this study, we validate a statistical approach to find stably expressed reference genes for any experimental setting using a given set of candidates. We compare our approach to RNA sequencing which quantifies the expression of thousands of genes at the same time. We highlight the advantages of our approach which is cost effective and saves a lot of time when compared to sequencing.

## Introduction

Normalization of relative gene expression using qPCR assays relies crucially on the use of reference genes that exhibit minimal variation across experimental conditions [1–5]. Thus, the choice of reference genes has a significant bearing on the normalized profiles of target genes. In the last two decades, many statistical approaches have been proposed to help researchers identify stable reference genes from a given set of candidates [6–11]. However, the calculations and assumptions employed in these approaches fundamentally differ and could give rise to conflicting results [12]. To address these concerns, we recently reviewed these approaches and following a comparison of existing methods including Coefficient of Variation (CV) analysis, GeNorm, NormFinder, pairwise ΔCT method and Best Keeper, we proposed an effective qPCR analysis workflow [12]. Our approach combining visual representation and statistical testing of intrinsic variation, CV analysis for identifying overall reference gene variation and the NormFinder algorithm, proved to be effective in determining stable reference genes from a given set of candidates.

However, more recently, a growing number of studies have proposed the use of bulk RNA-Seq to screen for stable reference genes followed by the application of existing statistical methods to obtain the "best" reference genes to normalize qPCR assays [13–21]. In these studies, the intrinsic validity of screening for reference genes directly from the bulk RNA-Seq data is clear, as they have used RNA-Seq to identify stable reference genes in an effort to standardize the use of certain genes for their experimental setting. Although bulk RNA-Seq data analysis pipelines quantify differential expression of target genes, performing a qPCR confirmation is often prudent as genes that exhibit shorter transcript lengths and lower expression levels exhibit discordant results between RNA-Seq and qPCR [22,23]. This is largely because frequently used RNA-Seq normalization strategies are prone to overall and sample specific transcript-length bias wherein longer transcripts are attributed with more counts regardless of expression levels [24,25]. Moreover, in a standard RNA-Seq experiment employing 3–4 biological replicates per condition, a vast majority of the reads arise from a small set of highly expressed genes and thus there is an inherent discrimination towards genes that are less expressed in the system [26,27]. These factors are mostly likely at play when discordant differential expression results are obtained between qPCR and RNA-Seq [23]. Thus, the selection of qPCR reference genes from RNA-Seq data is not the best strategy as it can potentially lead to sub-optimal candidates. Moreover, the cost-benefit trade off in performing RNA-Seq to identify stable reference genes for qPCR assays is negligible. This approach is also not always feasible, especially in cases of sample scarcity or when dealing with poor yields of RNA. It is

particularly relevant when we are interested in only a small number of target genes whose expression can be assessed solely by qPCR. However, although qPCR is not prone to the same biases as RNA-Seq, the strongest impediment to performing reliable qPCR data analysis is poor reference gene selection for a given experimental setting. Thus, the approach used to validate reference genes is of utmost importance and it should ideally not be dependent on a specific source of "stable" candidates. It is also advantageous if suitable reference genes can be identified by the same technology rather than RNA-Seq.

In this study, we address these concerns by demonstrating that the statistical approach employed to validate reference genes is more important than the preselection of stable genes from bulk RNA-Seq data. Specifically, with the right statistical approach, reference genes filtered from any conventional set of candidates generate the same differential expression results when compared to stable genes preselected from bulk RNA-Seq data. Furthermore, the results obtained are also comparable with the fold changes observed in RNA-Seq for the genes that we tested in qPCR. These results taken together demonstrate that the preselection of candidate reference genes is not of any significance for data normalization and hence, RNA-Seq data is not an essential requisite to obtain robust reference genes for qPCR data normalization.

## Materials and methods

### Ethics statement

All aspects of animal care and animal experimentation were performed in accordance with the relevant guidelines and regulations of INSERM, Université de Paris, and approved by the French National Committee of Animal experimentation and ethics.

### Sample procurement

**iPSC Microglia.** The CRISPR/Cas gene edited TREM2 knock-out iPSC line BIONi010-C-17 (TREM2 KO) and its isogenic control BIONi010-C (TREM2 WT) were purchased from EBISC and maintained at 37˚C 5% CO2 with E8-Flex medium. Microglia were differentiated from these lines as previously described [28]. In short, using AggreWell 800 (34850, StemCell Technologies), embryonic bodies (EBs) were prepared from iPSCs in E8-Flex medium supplemented with 50 ng/ml VEGF (PHC9394, ThermoFisher), 50 ng/ml BMP4 (120-05ET, Peprotech), and 20 ng/ml SCF (300–07, PeproTech) for 3 days with 75% medium change each day. EBs were transferred to a T75 flask in X-VIVO15 (BE02-060F, Lonza) medium supplemented with 25 ng/ml IL-3 (PHC0031, ThermoFisher), 2 mM Glutamax (35050061, ThermoFisher), 100 ng/ml M-CSF (300–25, PeproTech), and 0.055 mM β-mercaptoethanol (31350–010, ThermoFisher). Precursor cells started to emerge from the EBs approximately 4 weeks later, and were collected by gently tapping the flask. These precursor cells are differentiated to microglia over 7-days in DMEM F:12, Neuronal basal medium plus (1:1) supplemented with 100 ng/ml IL-34, 10 ng/ml GM-CSF, and 100 ng/ml M-CSF.

**Sciatic nerves.** To assess the stability of commonly used reference genes during myelination of the sciatic nerves, P3 and P21 C57BL6/J mice (3 Males and 4 females per time point) were dissected and the sciatic nerves were harvested for RNA extraction. The RNA-seq data for these time points was mined from a publicly available dataset as explained in the *"Bulk RNA-seq and data analysis"* section below.

### Total RNA extraction

**iPSC microglia.** Approximately $2x10^5$ iPSC derived microglial cells were harvested using TRIzol reagent (Ambion Life Technologies 15596018) for total RNA isolation. Total RNA was

isolated using Direct-Zol$^{TM}$ RNA microprep columns (Zymo research R2062) according to the manufacturer's protocol.

**Sciatic nerves.**    Total RNA was extracted from sciatic nerves using 1 mL of TRIzol reagent (Ambion Life Technologies 15596018) on ice using the manufacturer's instructions with slight modifications. Briefly, 100% ethanol was substituted for isopropanol to reduce the precipitation of salts. In addition, RNA precipitation was carried out overnight at -20˚C in the presence of glycogen (0.02mg/mL final concentration). The following day, precipitated RNA was pelleted by centrifugation and washed at least 3 times with 70% Ethanol to eliminate any residual contamination. Tubes were then spin-dried in a vacuum centrifuge for 5 minutes and RNA was resuspended in 20 μL of RNA resuspension buffer containing 0.1mM EDTA, pH 8. RNA was then stored at -80˚C till RTqPCR.

## RNA quality, integrity, and assay

**iPSC Microglia.**    RNA quantity was assayed using UV spectrophotometry on Nanodrop One (Thermo Scientific). Optical density absorption ratios A260/A280 & A260/A230 of the iPSC microglia samples were 2.0 (±0.1 SD) and 2.1 (±0.1 SD) respectively. RNA integrity was verified using Agilent bioanalyzer. All the samples exhibited a RIN score ≥ 9 and were subsequently used for downstream analysis including paired-end bulk RNA-seq and qPCR.

**Sciatic nerves.**    RNA quantity was assayed using UV spectrophotometry on Nanodrop One (Thermo Scientific). Optical density absorption ratios A260/A280 & A260/A230 of sciatic nerve samples were 1.84 (±0.04 SD) and 1.54 (±0.53 SD) respectively. RNA integrity was verified using Agilent bioanalyzer at the Genomics Platform at Institut Cochin, Paris. All samples exhibited a RIN score ≥ 8.8 and were used for qPCR analysis.

## Bulk RNA-seq and data analysis

For the iPSC samples, total RNA was extracted and sent to Genewiz, Germany for paired-end bulk RNA-seq. The raw data, final processed data and the metadata are available in the GEO database (GSE178924). For the sciatic nerves, bulk RNA-Seq data was obtained from a previously published study [29]. The raw data can be accessed at the Zenodo repository (https://zenodo.org/record/1154250). Only the WT datasets at P3 and P21 were mined and re-analyzed. Fastq files were aligned using the STAR algorithm (version 2.7.6a). Reads were then counted using RSEM (v1.3.1) and the statistical analyses on the read counts were performed with the DESeq2 package (DESeq2_1.28.1) to determine the proportion of differentially expressed genes between the two experimental conditions [30]. The standard DESeq2 normalization method (DESeq2's median of ratios with the DESeq function) was used, with a pre-filter of reads and genes (reads uniquely mapped on the genome, or up to 10 different loci with a count adjustment, and genes with at least 10 reads in at least 3 different samples). The biomaRt package (v2.44.4) was used to substitute gene names for the Ensembl IDs in the count matrix [31]. The DESeq2 pipeline was used to fit a generalized linear model (GLM) for the expression of each gene relative to the experimental groups. To determine the variation in gene expression, the dispersion value computed by the GLM was tabulated into the results data frame. Square root of dispersion was calculated using sqrt function from base functions in R (v4.0.2). The coefficient of dispersion (CVfromDisp) was calculated by multiplying square root of dispersion by 100.

## Reference gene selection from RNA-Seq

The data frame was subjected to 4 filtration methods to select stable genes in both the iPSC and Sciatic nerve datasets. First filtration was done by retaining all the genes with padj value

above 0.05. The resulting data frame was ordered based on the log2FoldChange (log2FC) column and the second filtration was performed by retaining all the genes with log2FC between -0.1 to 0.1 implying negligible intergroup variation. Third filtration was done by retaining all the genes that lie in the CVfromDisp range between 10 & 20. The fourth and last filtration was performed by retaining all the genes that exhibited a Basemean value greater than or equal to 500 implying sufficient expression for qPCR detection.

Once these criteria were applied, 683 genes were obtained (basemean values ranging from 500 to 123000) in the iPSC dataset and 42 genes were obtained (basemean values ranging from 500 to 48000) in the sciatic nerve dataset. To obtain stable reference genes with both low and high expression, the final list of candidates was partitioned into quartiles based on basemean expression. 3 reference genes from each of the first three quartiles and 1 reference gene from the last quartile were chosen; thereby generating 10 candidates. This selection strategy resulted in candidate reference genes that exhibit Cq values between 18 cycles (high expression) and 26 cycles (relatively lower expression) in qPCR.

## Conventional reference gene selection

All conventional reference genes chosen do not have any proven stability for the experimental setting in question. For the iPSC dataset, the conventional human reference genes chosen were *ACTB*, *GAPDH*, *GUSB*, *HPRT*, *PGK1*, *PPIA*, *RPL13A*, *TBP*, and *UBC*. For the sciatic nerve dataset, the conventional mouse reference genes chosen were *Actb*, *Gapdh*, *Tbp*, *Sdha*, *Pgk1*, *Ppia*, *Rpl13a*, *Hsp60*, *Mrpl10*, *Rps26*. These mouse reference genes have been previously used to establish the qPCR data analysis workflow [12]

## Target gene selection

Differentially expressed genes that exhibited Padj<0.05 were first retained from the results dataframe. Next, genes that exhibited log2FC values above +0.6 and below -0.6 were retained and partitioned into 2 separate lists. 3 target genes were randomly chosen from each of these lists for differential expression analysis in the iPSC dataset. In the Sciatic nerve dataset, 3 upregulated genes and 3 downregulated genes were chosen based on the recently published Sciatic Nerve Atlas https://www.snat.ethz.ch [32].

## Primer design

All primers used in the study were designed using the Primer 3 plus software (https://primer3plus.com/cgi-bin/dev/primer3plus.cgi). Splice variants and the protein-coding sequence of the genes were identified using the Ensembl database (www.ensembl.org). Constitutively expressed exons among all splice variants were then identified using the ExonMine database [33]. Primers that spanned two subsequent constitutively expressed exons were then designed using the Primer 3 plus software. The amplicon size of all primers was between 88bp– 200bp. For detailed information on Primer sequences refer to the metadata - https://doi.org/10.6084/m9.figshare.15169104.v2

## Amplification efficiencies

The amplification efficiencies of primers were calculated using serial dilution of cDNA molecules. Briefly, cDNA from both the experimental groups of the sciatic nerve (for mouse primers) and iPSC (for human primers) were serially diluted four times by a factor of 10 (1, 1:10, 1:100 & 1:1000). qPCR was then performed using these dilutions and the results were plotted as a standard curve against the respective concentration of cDNA. If the Cq values of the 4th

dilution fell beyond the detection range of the machine or closer to the No-RT Control, only the first three dilutions were taken into consideration. Amplification efficiency (E) was calculated by linear regression of standard curves using the following equation: $E = 10^{(-1/\text{Slope of the standard curve})}$. Primer pairs that exhibited an Amplification Efficiency (E) of 1.9 to 2.1 (95% - 105%) and an $R^2$ value (Determination Coefficient) of 0.99 or above were chosen for this study. The Cq values of all genes across different groups were well within the range of the standard dilution curve. Standard curves and amplifications efficiencies of all primers used can be accessed from the metadata - https://doi.org/10.6084/m9.figshare.15169104.v2

## RT-qPCR

**iPSC microglia.**   1000 ng of total RNA was first subjected to DNase digestion (Thermo scientific EN0525) at 37°C for 30 minutes to eliminate contaminating genomic DNA. Next, DNase activity was stopped using EDTA (Thermo scientific) and cDNA synthesis was done using iSCRIPT™ cDNA synthesis kit from BioRad (1708891) to a total volume of 20 μl. cDNA was diluted 1:10 with nuclease free water for qPCR analysis. qPCR was performed using Takyon ROX SYBR 2X MasterMix (Eurogentec UF-RSMT-B0701) as a fluorescent detection dye. All reactions were carried out in a final volume of 8 μl in 384 well plates with 300 nM gene-specific primers, around 5 ng of cDNA (at 100% RT efficiency) and 1X SYBR Master Mix in each well. Each reaction was performed in triplicates. All qPCR experiments were performed using ThermoFisher Scientific QuantStudio 7 with a No-Template-Control (NTC) to check for primer dimers and a No-RT-Control (NRT) to check for any genomic DNA contamination.

**Sciatic nerves.**   500 ng of total RNA was reverse transcribed with Random Primers (Promega C1181) and MMLV Reverse Transcriptase (Sigma M1302) according to prescribed protocols. qPCR was performed using Absolute SYBR ROX 2X qPCR mix (Thermo AB1162B) as a fluorescent detection dye. All reactions were carried out in a final volume of 7 μl in 384 well plates with 300 nM gene-specific primers, around 3.5 ng of cDNA (at 100% RT efficiency) and 1X SYBR Master Mix in each well. Each reaction was performed in triplicates. All qPCR experiments were performed on BioRad CFX384 with a No-Template-Control (NTC) to check for primer dimers and a No-RT-Control (NRT) to check for any genomic DNA contamination.

## qPCR statistical analysis and data visualization

qPCR readouts were analyzed in Precision Melt Analysis Software v1.2 (Sciatic nerve samples) and QuantStudio Real-Time PCR Software v1.7.1 (iPSC samples). The amplicons were subjected to Melt Curve analysis and were verified for a single dissociation peak at a Melting Temperature (Tm) > 75°C as expected from the primer constructs. The Cq data was exported to Microsoft Excel for further calculations. Each biological sample had three technical replicates thereby generating three individual Cq values. The arithmetic mean of the triplicates was taken to be the Cq representing the biological sample. The standard deviation (SD) of the triplicates was also calculated and samples that exhibited SD > 0.20 were considered inconsistent. In such cases, one outlier Cq was removed to have at least duplicate Cq values for each biological sample and an SD < 0.20.

Reference gene validation was performed according to our qPCR data analysis workflow [12]. Visual representation of potential intrinsic variation in reference genes was identified by plotting the raw expression profiles ($2^{-\Delta Cq}$) of all candidate reference genes. Due to reduced sample sizes, a non-parametric Mann Whitney U-Test was performed to assess statistically significant expression variation between experimental groups. The alpha value was set at 0.05 for statistical significance. The genes were then screened using Coefficient of variation (CV)

analysis to eliminate genes that exhibited CV>50% as they could impede the robustness of the Normfinder algorithm [12]. Normfinder was then used to determine the best pair of reference genes (least S value) to compute the Normalization Factor (NF) for qPCR data normalization [8]. From the resulting pair of stable reference genes, the NF was then calculated as the arithmetic mean of the Cq values of the 2 genes for each sample. Relative expression of target genes was then quantified using the $2^{-\Delta\Delta Ct}$ method and data was normalized by the NF calculated from the 2 best reference genes [34,35]. Relative fold change data was then analysed and visualised using R studio. Statistical tests used for comparing experimental groups are indicated in the respective Figure legends.

## Results

### Reference gene selection from RNA-Seq of microglia (WT vs TREM2KO)

A growing body of literature promotes the use of RNA-Seq to identify stable reference genes for qPCR data normalization in different experimental conditions. However, we believe that this approach is superfluous and that stable reference genes can readily be identified by starting out with any conventional set of candidates and adopting the right statistical approach to determine the best reference genes for a given experimental condition. To test this hypothesis, we first performed paired-end bulk RNA-Seq on iPSC-derived WT and TREM2KO microglia, an experimental model used to study the role of microglia in neuroinflammation and neurodegeneration [36,37]. Based on the criteria detailed in the Materials and Methods section (see *Reference gene selection from RNA-Seq*), we shortlisted 10 candidate reference genes (*ANXA7*, *APIP*, *CCNT2*, *CNBP*, *DDX42*, *FOXK1*, *KIF13A*, *MRPL37*, *PPP1R10*, and *USP5*) from our RNA-Seq data. The base mean values of these genes and other RNA-Seq data features are detailed in **S1 Table**.

### Reference gene validation of RNA-Seq derived reference genes (WT vs TREM2KO)

The reference genes selected from RNA-Seq were then subjected to validation using the qPCR workflow that we developed recently [12]. We first computed the intrinsic variation of reference genes by linearizing the Cq values ($2^{-Cq}$) followed by visually representing the non-normalized expression levels ($2^{-\Delta Cq}$) of the genes in both experimental groups (**Fig 1**). The WT group was used as the experimental calibrator. Subsequently, statistical testing was performed using a non-parametric Mann Whitney Test to determine significant differences between the groups. Among the 10 genes tested, only 2 genes (MRPL37 and PPP1R10) showed significant variation in non-normalized expression levels between the WT and TREM2KO groups (**Fig 1**).

In accordance with our workflow, we next performed Coefficient of Variation analysis on linearized Cq values ($2^{-Cq}$) to assess the overall variation (both groups included) of the reference genes (**Table 1**). The genes tested exhibited CV values between 5.99% and 15.75% and they were ranked as shown in **Table 1**. As all genes exhibited CV<50%, they could therefore be screened using the NormFinder algorithm to determine the best combination of reference genes to be used for data normalization. This is a crucial step of reference gene validation as genes that exhibit CV>50% can compromise the robustness of the algorithm [12]. NormFinder determines the stability of a reference gene by factoring the intergroup as well as the intragroup variation and computes a Stability S Score [8]. The lower the S score, the higher the stability of the gene across all experimental groups. The algorithm also suggests the best pair of reference genes that can be used for robust data normalization. NormFinder results of the selected reference genes from RNA-seq are represented in **Table 1** along with their respective

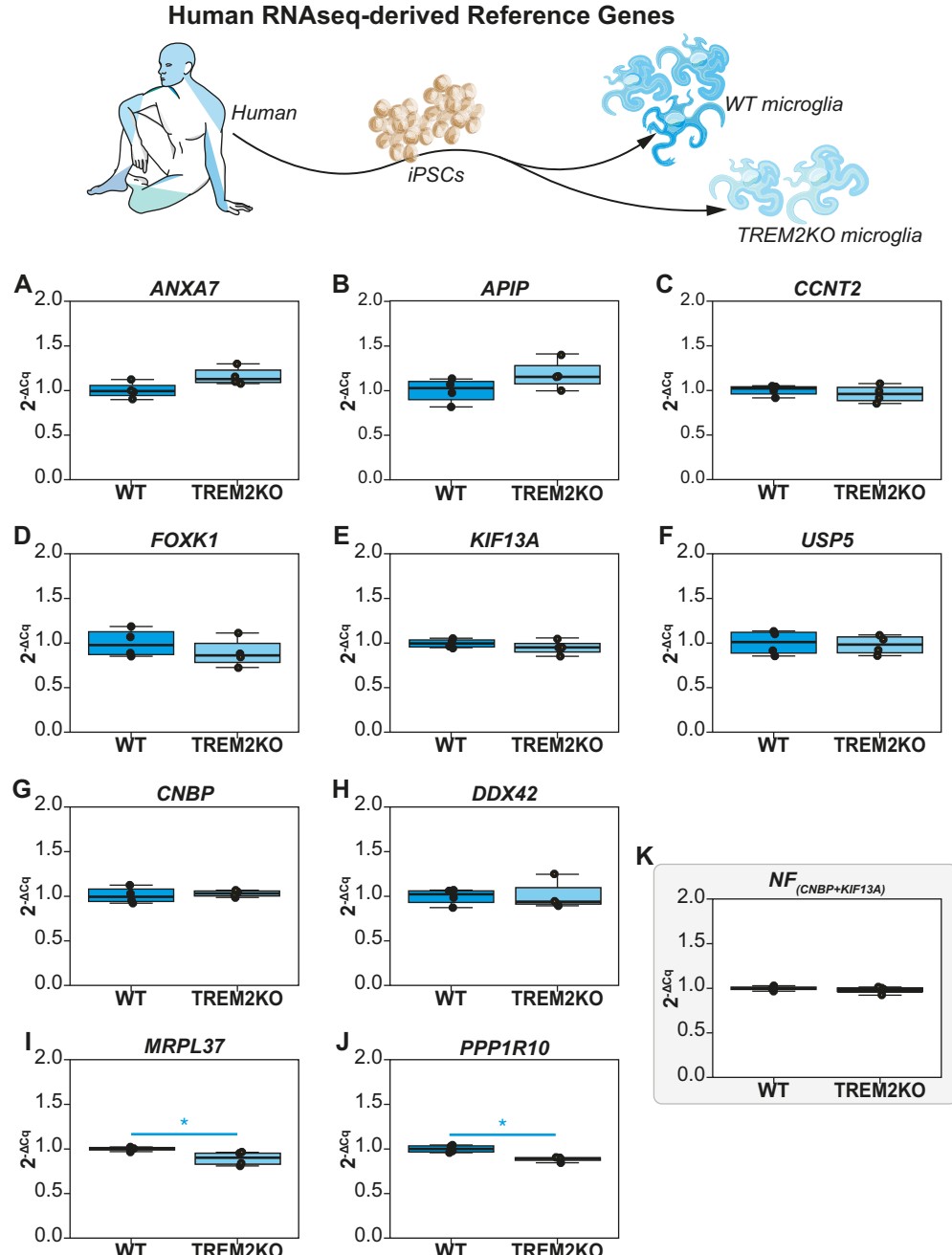

**Fig 1. Non-normalized expression profiles (2^−ΔCq) of reference genes derived from RNA-Seq data of iPSC derived microglia.** WT group was used as the experimental calibrator. (A) ANXA7, (B) APIP, (C) CCNT2, (D) FOXK1, (E) KIF13A, (F) USP5, (G) CNBP, (H) DDX42, (I) MRPL37, (J) PPP1R10 and (K) Normalization Factor (NF) assessed by NormFinder that combines CNBP & KIF13A. Non-parametric Mann Whitney U test was used to assess differences between the groups. The alpha value was set at 0.05 and P values are annotated as follows: * P<0.05.

S Scores and ranks. The algorithm further revealed CNBP and KIF13A to be the best combination of genes with a grouped Stability S score of 0.05 (**Table 1**). The Cq values of these two genes were then combined to compute the Normalization Factor (NF). Evidently and as expected, the non-normalized expression level of the NF does not vary significantly between

**Table 1. CV and NormFinder analysis RNA-Seq derived reference genes in iPSC microglia.** Expression stability of candidate reference genes derived from bulk RNA-Seq data were evaluated using Coefficient of Variation (CV) Analysis & NormFinder.

| CV Analysis | | | NormFinder | | |
|---|---|---|---|---|---|
| Gene | %CV | Rank | Gene | Stability S | Rank |
| CNBP | 5.99 | 1 | CNBP | 0.07 | 1 |
| KIF13A | 6.36 | 2 | CCNT2 | 0.08 | 2 |
| PPP1R10 | 6.87 | 3 | KIF13A | 0.1 | 3 |
| CCNT2 | 7.46 | 4 | PPP1R10 | 0.1 | 4 |
| MRPL37 | 7.74 | 5 | USP5 | 0.1 | 5 |
| ANXA7 | 10.58 | 6 | MRPL37 | 0.11 | 6 |
| USP5 | 10.81 | 7 | DDX42 | 0.12 | 7 |
| DDX42 | 11.41 | 8 | ANXA7 | 0.15 | 8 |
| APIP | 14.61 | 9 | FOXK1 | 0.15 | 9 |
| FOXK1 | 15.75 | 10 | APIP | 0.16 | 10 |

Normfinder Best Pair: CNBP/KIF13A

Grouped Stability: 0.05

the two groups (**Fig 1K**). It is to be noted that the combination of the two best reference genes exhibits lesser intergroup and intragroup variation (**Fig 1K** **versus the rest**) including the top ranked CNBP (**Fig 1G**). This is an inherent feature of the algorithm.

## Reference gene validation of conventional human reference genes (WT vs TREM2KO)

In addition to the reference genes derived from the RNA-Seq data, we next sought to validate commonly used human reference genes chosen arbitrarily. We selected 10 reference genes (*ACTB*, *GAPDH*, *GUSB*, *PPIA*, *RPLP0*, *TBP*, *HPRT*, *PGK1*, *UBC*, *and RPL13A*) and assessed their intergroup variation by plotting their non-normalized profiles (**Fig 2**). This was followed by statistical testing for significant differences between the two experimental groups. Among the 10 genes tested, none of the genes exhibited significant variation in their non-normalized expression levels (**Fig 2**).

These genes were then subjected to CV analysis to assess the overall variation (**Table 2**). They exhibited CV values ranging from 5.29% to 32.37%. However, all genes satisfied the CV<50% criterion for NormFinder screening. NormFinder results for these genes are also shown in **Table 2**. Interestingly and in contrast to the reference genes derived from the RNA-Seq data (**Table 1**), the Stability S score of the conventional genes are largely lower than the genes derived from RNA-Seq, thus indicating better stability. The first 6 ranks from RPL13A to GAPDH have S scores less than 0.1 whereas in the previous dataset (**Table 1**), only the first 2 ranks (CNBP & CCNT2) exhibit scores less than 0.1. Finally, among all the conventional reference genes screened, NormFinder suggested the use of RPLP0 and GUSB as the best pair for data normalization with a grouped stability of 0.03 (**Table 2**); which in comparison, is lesser than the Stability Score of 0.05 from CNBP and KIF13A (**Table 1**). In principle, these results show that conventional reference genes could potentially exhibit better stability than stable reference genes filtered from RNA-Seq data.

## Differential expression of target genes by qPCR and RNA-Seq (WT vs TREM2KO)

Following reference gene validation of conventional and RNA-Seq-derived candidates, we picked 6 target genes that were differentially expressed based on the RNA-Seq results (*see*

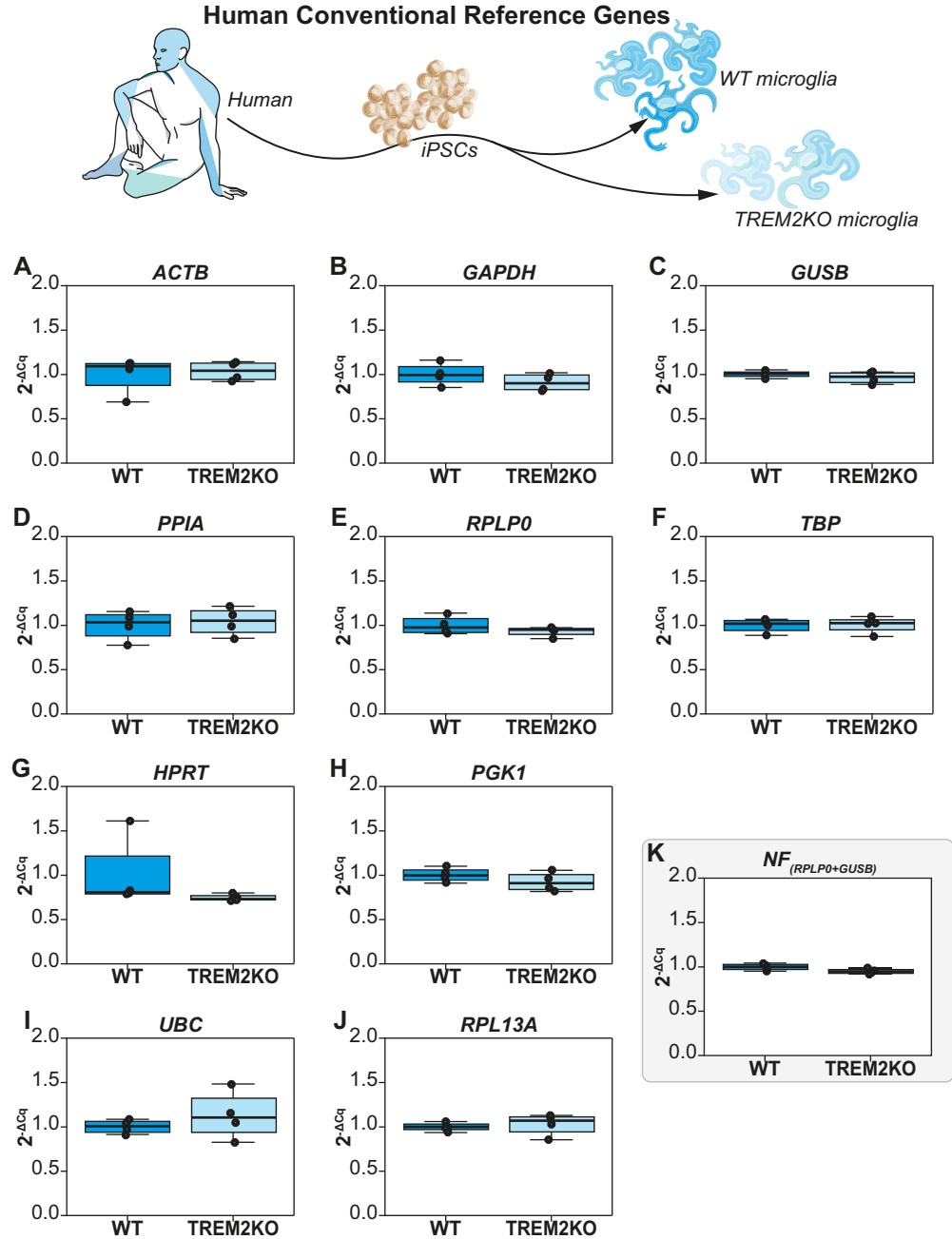

**Fig 2. Non-normalized expression profiles (2$^{-\Delta Cq}$) of conventional human reference genes in iPSC derived microglia.** WT group is used as the experimental calibrator. (A) ACTB, (B) GAPDH, (C) GUSB, (D) PPIA, (E) RPLP0, (F) TBP, (G) HPRT, (H) PGK1, (I) UBC, (J) RPL13A, and (K) Normalization Factor (NF) assessed by NormFinder that combines RPLP0 & GUSB. Non-parametric Mann Whitney U test was used to assess differences between the groups. The alpha value was set at 0.05 and P values are annotated as follows: * $P < 0.05$, ** $P < 0.01$, *** $P < 0.001$.

*Target gene selection* from *Materials & Methods*). We included 3 target genes (*TREM2*, *P2RY12* and *CD52*) that were significantly downregulated and 3 others (*ACSS3*, *SYDE1* and *TBX10*) that were significantly upregulated. The relative expression of these genes was then computed by qPCR using the Normalization Factor (NF) from conventional candidate reference genes (*RPLP0* & *GUSB* or Conventional NF) or from RNA-Seq-derived candidate

**Table 2. CV and NormFinder analysis of conventional reference genes iPSC microglia.** Expression stability of conventional candidate reference genes were evaluated using Coefficient of Variation (CV) Analysis, NormFinder.

| CV Analysis | | | NormFinder | | |
|---|---|---|---|---|---|
| Gene | %CV | Rank | Gene | Stability S | Rank |
| GUSB | 5.29 | 1 | RPL13A | 0.03 | 1 |
| TBP | 7.56 | 2 | GUSB | 0.04 | 2 |
| RPL13A | 8.16 | 3 | RPLP0 | 0.05 | 3 |
| RPLP0 | 8.18 | 4 | TBP | 0.05 | 4 |
| PGK1 | 9.56 | 5 | PGK1 | 0.06 | 5 |
| GAPDH | 11.35 | 6 | GAPDH | 0.07 | 6 |
| PPIA | 13.75 | 7 | PPIA | 0.10 | 7 |
| ACTB | 14.31 | 8 | UBC | 0.11 | 8 |
| UBC | 17.47 | 9 | ACTB | 0.14 | 9 |
| HPRT | 32.27 | 10 | HPRT | 0.18 | 10 |

NormFinder Best Pair: RPLP0/GUSB

Grouped Stability S: 0.03

reference genes (*CNBP* & *KIF13A* or RNA-Seq derived NF) (**Fig 3**). To compare our qPCR results, we also computed the fold change assessed by RNA-Seq using the Normalized counts of these genes.

Regarding the downregulated genes, *TREM2* expression in the TREM2KO group, when assessed through qPCRs, exhibited expression fold change values of $0.31 \pm 0.05$ (mean FC ± SD, conventional NF, **Fig 3A**) and $0.35 \pm 0.04$ (RNA-Seq derived NF, **Fig 3B**). In RNA-Seq, the fold change values were $0.23 \pm 0.03$ (**Fig 3C**). *P2RY12* fold change values in the TREM2KO group were $0.04 \pm 0.02$ when assessed using qPCRs (both conventional NF and RNA-Seq derived NF, **Fig 3D and 3E**). However, RNA-Seq fold change values of *P2RY12* were comparatively higher at $0.62 \pm 0.19$ (**Fig 3F**). *CD52* fold change in the TREM2KO group when assessed by qPCR were $0.38 \pm 0.06$ (Conventional NF, **Fig 3G**) and $0.40 \pm 0.05$ (RNA-Seq derived NF, **Fig 3H**). RNA-Seq fold changes were comparatively lower at $0.16 \pm 0.01$ (**Fig 3I**).

For upregulated genes, *ACSS3* in the TREM2KO group exhibited fold change values of $3.72 \pm 0.79$ (Conventional NF, **Fig 3J**) and $3.40 \pm 0.88$ (RNA-Seq derived NF, **Fig 3K**). The fold changes in RNA-Seq were $4.54 \pm 1.25$ (**Fig 3L**). *SYDE1* fold change values assessed by qPCR were at $3.58 \pm 1.18$ (Conventional NF, **Fig 3M**) and $3.29 \pm 1.28$ (RNA-Seq derived NF, **Fig 3N**). RNA-Seq expression levels of *SYDE1* were at $3.62 \pm 0.34$ (**Fig 3O**). Finally, *TBX10* expression fold changes assessed by qPCR were at $3.06 \pm 1.51$ (Conventional NF, **Fig 3P**) and $2.82 \pm 1.60$ (RNA-Seq derived NF, **Fig 3Q**). RNA-Seq fold changes were at $3.24 \pm 0.61$ (**Fig 3R**). These results taken together with the visual representation of differential expression described in **Fig 3** show that qPCR normalization using conventional NF or RNA-Seq derived NF render the same results. They do not always concur with RNA-Seq fold changes in magnitude but do so in tendency.

## Comparison of fold changes between qPCRs and RNA-Seq (WT vs TREM2KO)

We next investigated in detail if the Fold Change distributions in the TREM2KO experimental group computed across the 3 methods differed significantly from one another. To this end, we compared the distributions by a non-parametric ANOVA (Kruskal Wallis Test) of ordinal distributions followed by Dunn's multiple comparison post-test (**S1 Fig**). qPCR Fold changes

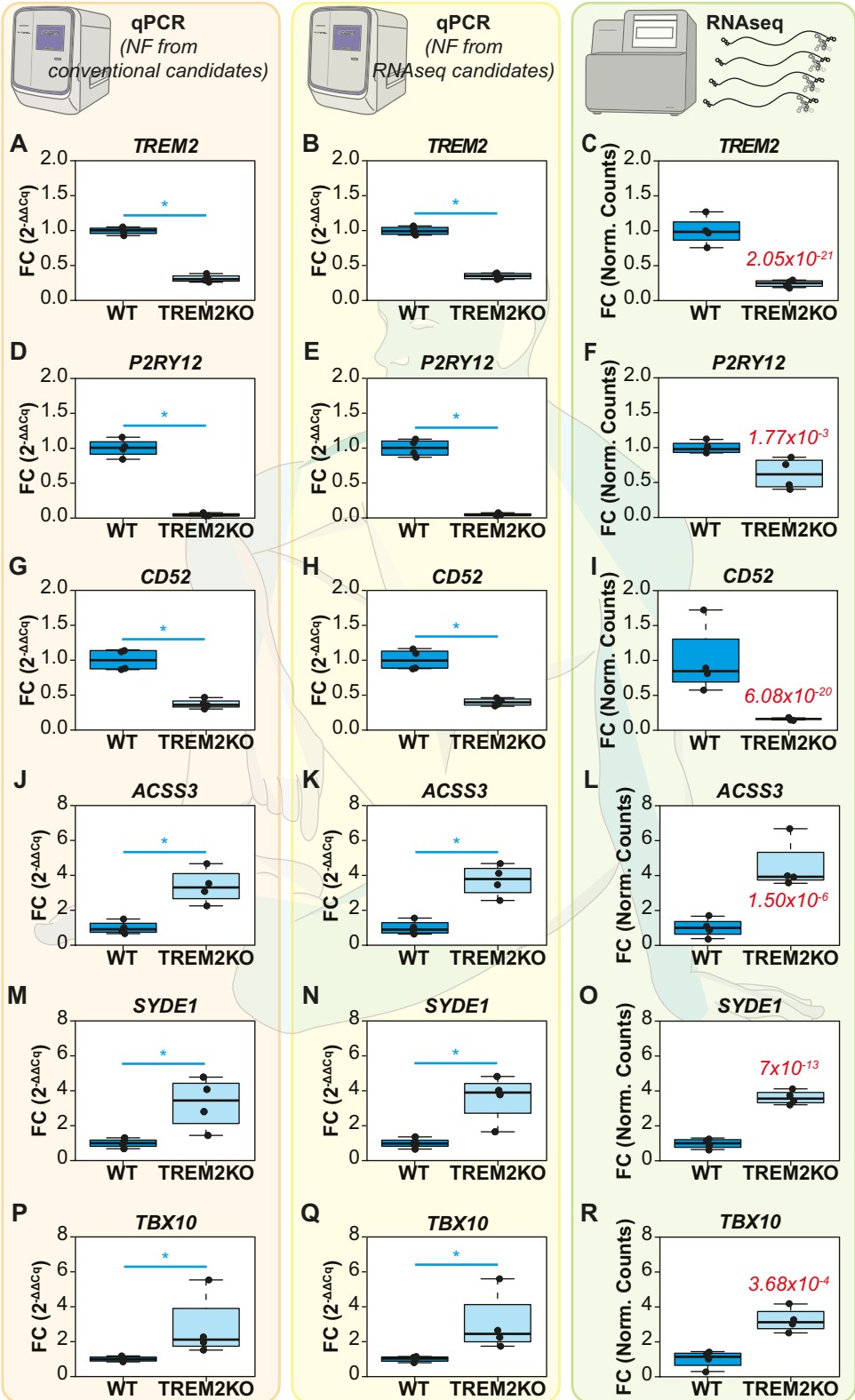

**Fig 3. Differential expression of target genes between WT and TREM2KO iPSC cells assessed by qPCRs and RNA-Seq.** The target genes (TREM2, P2RY12, CD52, ACSS3, SYDE1, TBX10) have been normalized with the normalization factor (NF) computed from conventional reference genes RPLP0 & GUSB in the first column (A, D, G,

J, M, P) or from RNA-Seq derived reference genes CNBP & KIF13A in the second column (B, E, H, K, N, Q). The third column contains the Fold Changes computed from RNA-Seq (C, F, I, L, O, R). The normalized counts for each gene in each group were used to calculate the fold changes in RNA-Seq. The WT group was used as the experimental control. The $P_{adj}$ value of RNA-Seq is indicated in the TREM2KO group for each gene in the RNA-Seq column. For the qPCRs, Non-parametric Mann Whitney U test was used to assess differences between the groups. The alpha value was set at 0.05 and P values are annotated as follows: * P<0.05.

computed using Conventional NF were used as the control for multiple comparisons. Therefore, the first comparison was made between the conventional and RNA-Seq derived NF groups (difference in qPCRs, hereafter referred to as Comparison 1) and the second comparison was made between the Conventional NF group and RNA-Seq differential expression (difference between qPCR and RNA-Seq, hereafter referred to as Comparison 2).

The WT experimental group was omitted from this analysis as it was used as the experimental calibrator for differential expression. The mean Fold Change of the WT group is always at 1 regardless of the gene/method in question and therefore it is redundant to test for statistical significance of the WT fold change levels across different methods for each gene.

Non-parametric ANOVA revealed no significant difference between the FC distributions of the upregulated genes *ACSS3*, *SYDE1* and *TBX1* (**S1D, S1E and S1F Fig**) suggesting that the qPCR fold changes concur with RNA-Seq fold changes in these genes. As for the downregulated genes, *TREM2* FC distributions varied significantly between the groups (Kruskal Wallis P < 0.5); however, Dunn's post test revealed no significant difference in Comparison 1 and Comparison 2 (**S1A Fig**). *P2RY12* FC distributions varied significantly between the groups (Kruskal Wallis P<0.01). Multiple comparisons revealed no difference in Comparison 1 but a significant difference in Comparison 2 (Dunn's Post Test P<0.05) (**S1B Fig**) CD*52* FC distributions varied significantly between the groups (Kruskal Wallis P < 0.5). Multiple comparisons, yet again, revealed no difference in Comparison 1 but a significant difference in Comparison 2 (Dunn's Post Test P<0.05) (**S1C Fig**). These results show firstly that no significant difference exists between the FC distributions of all the genes between the qPCRs (Comparison 1). Secondly, FC distributions of qPCRs and RNA-Seq do not show any difference for the upregulated genes *ACSS3*, *SYDE1* and *TBX10* (Comparison 2). However, for the downregulated genes, although no differences exist between qPCRs (Comparison 1); *P2RY12* & *CD52* FC distributions differ significantly between qPCR and RNA-Seq. In summation, these results prove beyond reasonable doubt that differential expression assessed by qPCR using conventional reference genes and "stable" reference genes filtered from RNA-Seq render similar expression profiles of target genes without any significant difference among the two normalization strategies. However, concordance with RNA-Seq results was only observed in the upregulated genes and could not be observed in the downregulated genes.

## Reference gene selection from RNA-Seq of mouse sciatic nerves (P3 vs P21)

To rule out any circumstantial evidence that may support our observations in the iPSC dataset, we sought to validate our results using a completely different experimental setting. We wanted to rule out the possibility that the conventional reference genes that we chose in the iPSC dataset were indeed stable reference genes by mere happenstance and thus they rendered the same profiles of differential expression as the reference genes filtered from RNA-Seq.

Therefore, in a confirmatory experimental approach we mined and re-analyzed RNA-Seq data that was previously generated from mouse sciatic nerves at Postnatal day 3 (P3) and P21 [29]. This dataset permits us to study RNA expression changes in the sciatic nerves during postnatal myelination. We applied the same criteria to select 10 candidate reference genes

(*Ppp3ca*, *Fkbp4*, *Vcp*, *Lama2*, *Ank2*, *Coq9*, *Chmp2a*, *Laptm5*, *Leprotl1*, *Supt4a*). The basemean values of these genes and other RNA-Seq data features are detailed in **S2 Table**.

### Reference gene validation of RNA-Seq derived reference genes from mouse sciatic nerves (P3 vs P21)

We evaluated the intrinsic intergroup variation of these reference genes by visual representation of their non-normalized profiles ($2^{-\Delta Cq}$) using P3 samples as the calibrator (**Fig 4**). As previously described, statistical inference was then assessed using the non-parametric Mann Whitney U test owing to reduced sample sizes. Interestingly, and in contrast to the iPSC dataset, 6 out of 10 genes (*Fkbp4*, *Lama2*, *Coq9*, *Chmp2a*, *Leprotl1 & Supt4a*) exhibit significant intergroup variation across the two experimental groups (**Fig 4A, 4B, 4C, 4D, 4E and 4F**).

   We next performed the CV analysis to assess overall variation (**Table 3**). The genes tested exhibited CV values between 12.32% and 36.24% and they were ranked as shown in **Table 3**. As all genes exhibited CV<50%, they could therefore be screened using the NormFinder algorithm. NormFinder analysis further revealed *Ppp3ca* and *Coq9* to be the best combination of genes with a grouped Stability S score of 0.04 (**Table 3**). Although Coq9 exhibited significant intergroup variation (**Fig 4E**), the combination of the gene with *Ppp3ca* resulting in the Normalization Factor (NF) does not exhibit any (**Fig 4K**). Interestingly, the RNA-Seq data features of the 6 genes that exhibit significant intergroup variation (**Fig 4A, 4B, 4C, 4D, 4E and 4F**) show that they are not differentially expressed. However, if these genes were to be normalized using the NF computed herein, some of them could potentially exhibit differential expression in qPCR as the NF is stable across experimental groups. It is a striking example to demonstrate the discordance of RNA-Seq and qPCR results in this dataset. However, the computation of a stable NF despite the presence of genes with significant intrinsic variation confirms the strength of the NormFinder algorithm and the validity of using the two best reference genes proposed by the method.

### Reference gene validation of conventional mouse reference genes (P3 vs P21)

To obtain stable reference genes from a set of conventional genes, we used 10 candidate mouse reference genes that we had previously used to establish our workflow (*Actb*, *Gapdh*, *Tbp*, *Sdha*, *Pgk1*, *Ppia*, *Rpl13a*, *Hsp60*, *Mrpl10*, *Rps26*) [12]. As explained above, visual representation followed by inferential statistics were performed and the results are described in **Fig 5**. 4 out of 10 genes (*Gapdh*, *Pgk1*, *Rps26 and Sdha*) exhibited significant intrinsic variation between the experimental groups (**Fig 5G, 5H, 5I and 5J**)

   CV analysis was then performed on these genes and the results are depicted in **Table 4**. CV values ranged from 12.3% to 30.44% and the genes were ranked accordingly. As all genes exhibited CV<50%, we screened these genes through the NormFinder algorithm to determine the best combination of reference genes for qPCR normalization. NormFinder determined *Hsp60* and *Ppia* to be the best combination of reference genes with a grouped Stability S Score of 0.06 (**Table 4**). Both *Hsp60* (**Fig 5C**) & *Ppia* (**Fig 5D**) did not exhibit any intergroup variation and the same was true for the NF computed by combining these genes (**Fig 5K**).

### Differential expression of target genes by qPCR and RNA-Seq (P3 vs P21)

Following reference gene validation of conventional and RNAseq-derived candidates, we picked 6 target genes that were known to be differentially expressed during postnatal development of the sciatic nerve [32]. 3 target genes (*Mpz*, *Mbp and Cd90*) that were chosen are

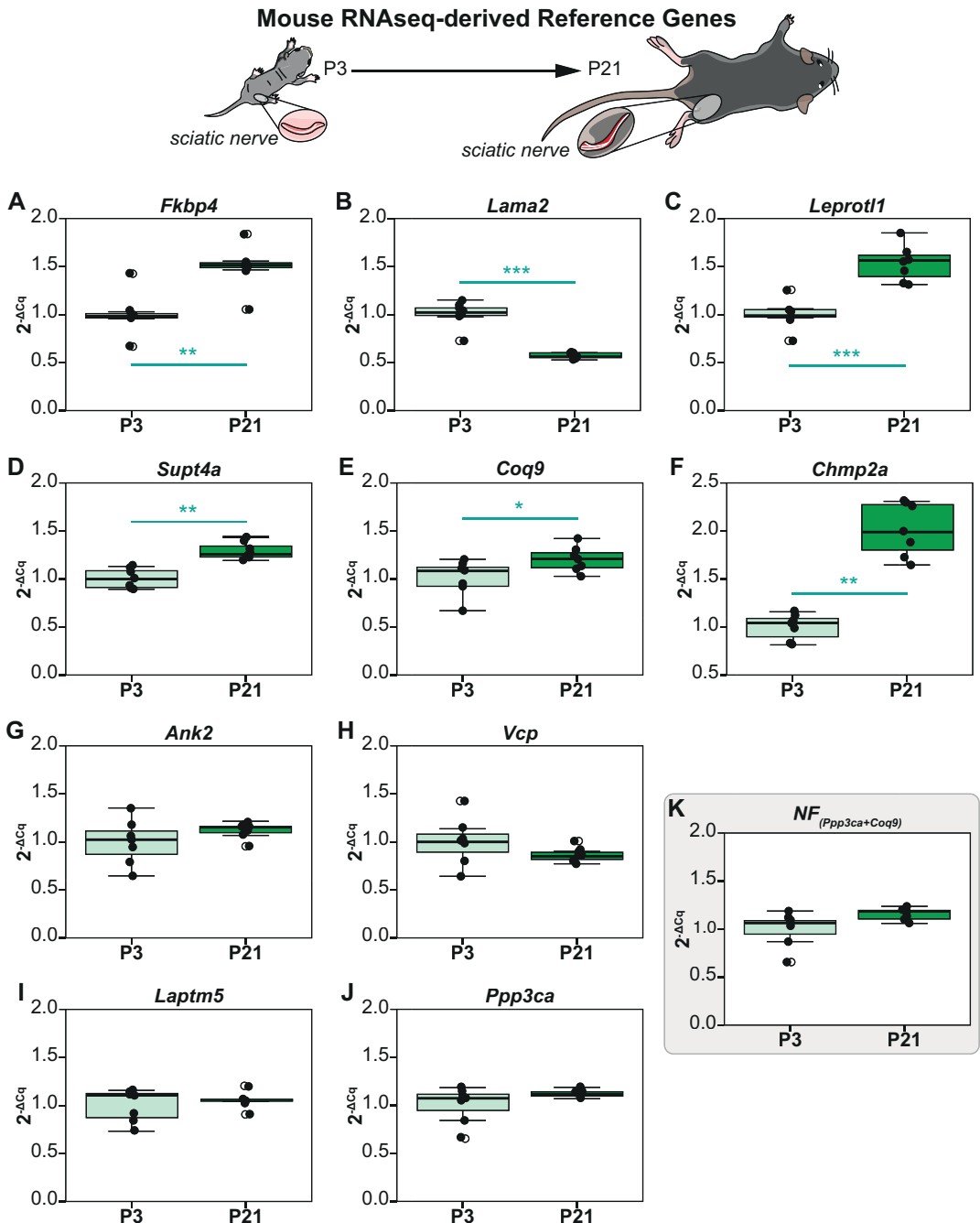

**Fig 4. Non-normalized expression profiles ($2^{-\Delta Cq}$) of reference genes derived from RNA-Seq analysis of sciatic nerves at post-natal day 3 (P3) and 21 (P21).** P3 group is the experimental calibrator. (A) *Fkbp4*, (B) *Lama2*, (C) *Leprotl1*, (D) *Supt4a*, (E) *Coq9*, (F) *Chmp2a*, (G) *Ank2*, (H) *Vcp*, (I) *Laptm5*, (J) *Ppp3ca*, and (K) Normalisation Factor (*Ppp3ca + Coq9*). *Non-parametric Mann Whitney U test was used to assess differences between the groups. The alpha value was set at 0.05 and P values are annotated as follows:* * *P<0.05,* ** *P<0.01,* *** *P<0.001.*

significantly upregulated and the other 3 target genes (*p75NTR*, *Mki67* and *Sox2*) are significantly downregulated at these time points. The relative expression of these genes was then computed by qPCR using the Normalization Factor (NF) from conventional candidate reference genes (*Hsp60* & *Ppia* or Conventional NF) or from RNAseq-derived candidate reference

**Table 3. CV and NormFinder analysis of reference genes derived from RNA-Seq in sciatic nerves.**

| CV Analysis | | | NormFinder | | |
|---|---|---|---|---|---|
| Gene | %CV | Rank | Gene | Stability S | Rank |
| *Laptm5* | 12.32 | 1 | *Ank2* | 0.07 | 1 |
| *Ppp3ca* | 12.96 | 2 | *Ppp3ca* | 0.08 | 2 |
| *Supt4a* | 14.79 | 3 | *Coq9* | 0.08 | 3 |
| *Coq9* | 16.26 | 4 | *Laptm5* | 0.16 | 4 |
| *Ank2* | 16.31 | 5 | *Supt4a* | 0.16 | 5 |
| *Vcp* | 19.59 | 6 | *Leprotl1* | 0.23 | 6 |
| *Leprotl1* | 24.49 | 7 | *Fkbp4* | 0.26 | 7 |
| *Fkbp4* | 25.86 | 8 | *Vcp* | 0.27 | 8 |
| *Lama2* | 29.96 | 9 | *Chmp2a* | 0.47 | 9 |
| *Chmp2a* | 36.24 | 10 | *Lama2* | 0.56 | 10 |

Normfinder Best combination: Ppp3ca + Coq9
Grouped Stability: 0.04

genes (*Ppp3ca* & *Coq9* or RNAseq derived NF) (**Fig 6**). To compare our qPCR results, we also computed the fold change assessed by RNA-Seq using the Normalized counts of these genes.

Regarding the downregulated genes, *Mki67* expression in the P21 group, when assessed through qPCRs, exhibited fold change values of 0.14 ± 0.10 (mean FC ± SD, conventional NF, **Fig 6A**) and 0.12 ± 0.08 (RNA-Seq derived NF, **Fig 6B**). In RNA-Seq, the fold change values were 0.08 ± 0.01 (**Fig 6C**). *p75NTR* expression fold change in the P21 group were 0.30 ± 0.04 (conventional NF, **Fig 6D**) and 0.26 ± 0.02 (RNA-Seq derived NF, **Fig 6E**) when assessed using qPCRs. However, RNA-Seq fold change values of *p75NTR* were comparatively higher at 0.63 ± 0.18 (**Fig 6F**). *Sox2* expression fold change in the P21 group when assessed by qPCR were 0.49 ± 0.05 (Conventional NF, **Fig 6G**) and 0.42 ± 0.03 (RNA-Seq derived NF, **Fig 6H**). RNA-Seq changes were at 0.39 ± 0.09 (**Fig 6I**).

For the upregulated genes, *Mbp* expression levels in the P21 group exhibited fold change values of 1.73 ± 0.15 (Conventional NF, **Fig 6J**) and 1.49 ± 0.19s (RNA-Seq derived NF, **Fig 6K**). The fold changes in RNA-Seq were 2.02 ± 0.62 (**Fig 6L**). *Mpz* expression levels assessed by qPCR were at 1.76 ± 0.38 (Conventional NF, **Fig 6M**) and 1.53 ± 0.35 (RNA-Seq derived NF, **Fig 6N**). However, RNA-Seq expression fold change of *Mpz* were significantly higher at 3.49 ± 0.26 (**Fig 6O**). Finally, *Cd90* expression fold change assessed by qPCR were at 1.83 ± 0.35 (Conventional NF, **Fig 6P**) and 1.57 ± 0.30 (RNA-Seq derived NF, **Fig 6Q**). RNA-Seq fold changes were at 4.81 ± 2.77 (**Fig 6R**). Yet again, these results taken together with the visual representation of differential expression described in **Fig 6** show that qPCR normalization using conventional NF or RNA-Seq derived NF render very similar results. We observe again that they do not always concur with RNA-Seq fold changes in magnitude, but they do so in tendency.

## Comparison of differential expression between qPCRs and RNA-Seq (P3 vs P21)

Similar to the analysis performed in the iPSC dataset, we next investigated in detail if the Fold Change distributions in the P21 experimental group computed across the 3 methods differed significantly from one another. We used non-parametric ANOVA (Kruskal Wallis Test) of ordinal distributions followed by Dunn's multiple comparison post-test (**S2 Fig**). As shown in the iPSC dataset, we performed multiple comparisons between qPCRs (Comparison 1) and also between qPCR and RNA-Seq (Comparison 2).

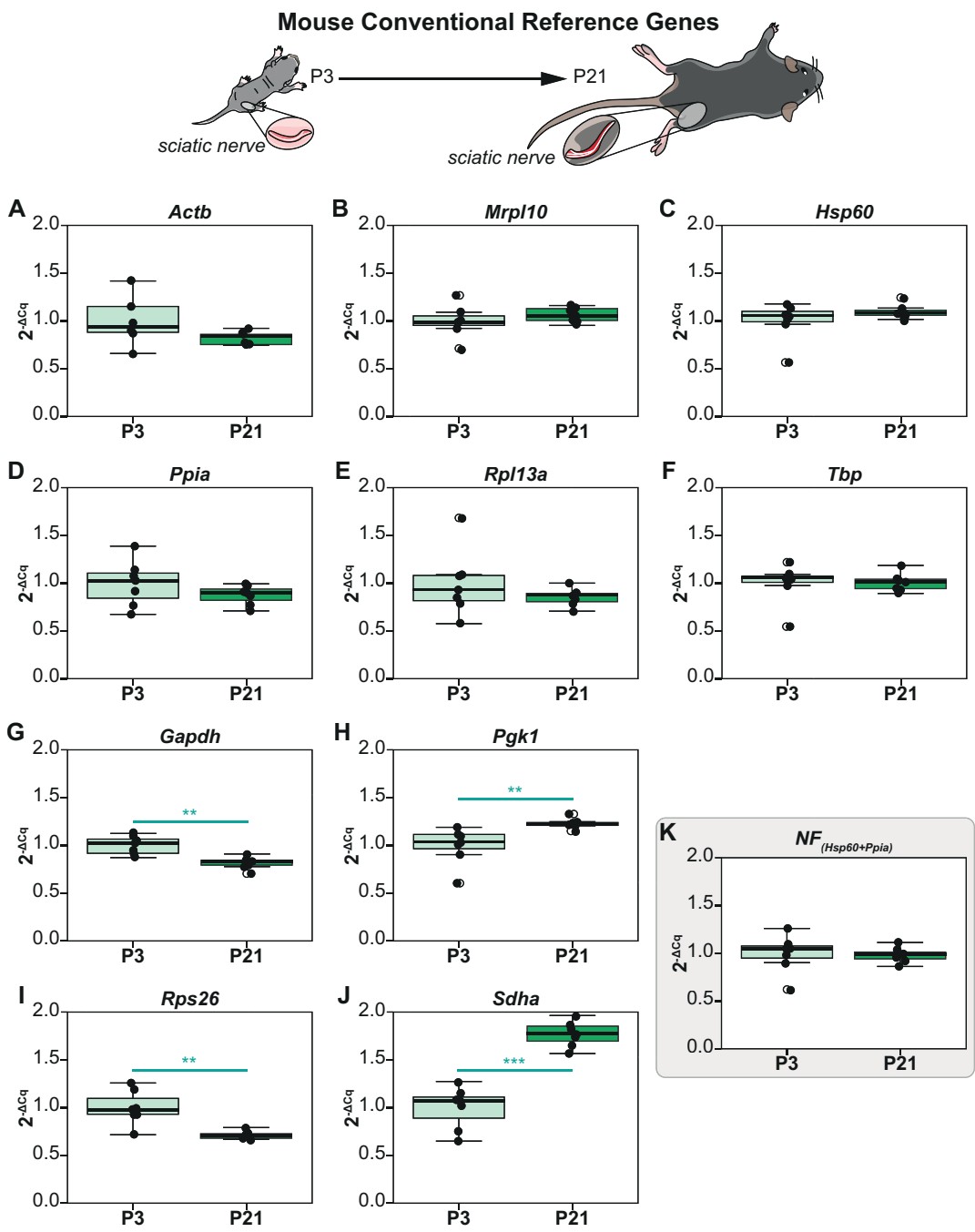

**Fig 5. Non-normalized expression profiles (2^{-ΔCq}) of conventional reference genes of sciatic nerves at post-natal day 3 (P3) and 21 (P21).** P3 group is the experimental calibrator. (A) Actb, (B) Mrpl10, (C) Hsp60, (D) Ppia, (E) Rpl13a, (F) Tbp, (G) Gapdh, (H) Pgk1, (I) Rps26, (J) Sdha, and (K) Normalisation Factor (Hsp60 + Ppia). Non-parametric Mann Whitney U test was used to assess differences between the groups. The alpha value was set at 0.05 and P values are annotated as follows: * P<0.05, ** P<0.01, *** P<0.001.

Non-parametric ANOVA revealed no significant difference between the FC distributions of *Mki67* and *Mbp* (**S2A and S2E Fig**) suggesting that the qPCR fold changes concur with RNA-Seq fold changes in these genes. *Sox2*, *p75NTR* and *Mpz* FC distributions varied significantly between the groups (Kruskal Wallis P < 0.05, P< 0.01 and P<0.01 respectively);

**Table 4. CV and NormFinder analysis of conventional reference genes in sciatic nerves.**

| CV Analysis | | | NormFinder | | |
|---|---|---|---|---|---|
| Gene | %CV | Rank | Gene | Stability S | Rank |
| Mrpl10 | 12.30 | 1 | Tbp | 0.03 | 1 |
| Gapdh | 13.06 | 2 | Mrpl10 | 0.04 | 2 |
| Hsp60 | 14.36 | 3 | Hsp60 | 0.05 | 3 |
| Tbp | 15.24 | 4 | Ppia | 0.05 | 4 |
| Pgk1 | 15.83 | 5 | Pgk1 | 0.06 | 5 |
| Ppia | 19.36 | 6 | Actb | 0.07 | 6 |
| Actb | 21.07 | 7 | Rpl13a | 0.10 | 7 |
| Rps26 | 22.04 | 8 | Gapdh | 0.11 | 8 |
| Rpl13a | 26.78 | 9 | Rps26 | 0.14 | 9 |
| Sdha | 30.44 | 10 | Sdha | 0.18 | 10 |

Normfinder Best combination: Hsp60 + Ppia
Grouped stability: 0.06

however, Dunn's post test revealed no significant difference in both Comparison 1 and Comparison 2 (**S2B, S2C and S2F Fig**). *Cd90* FC distributions varied significantly between the groups (Kruskal Wallis P<0.01). Multiple comparisons revealed no difference in Comparison 1 and Comparison 2 (**S2D Fig**). In summation, all the genes tested in the sciatic nerve dataset show no significant differences between Conventional NF and RNA-Seq-derived NF results. In summation, these results show once again that stable reference genes filtered from RNA-Seq data do not give any added advantage to qPCR data normalization and that not all genes exhibit comparable Fold Changes when comparing qPCR results and RNA-Seq.

## Discussion

The central premise of this article was to demonstrate that stable reference genes for qPCR data normalisation can be obtained from any conventional set of candidates provided the statistical approach of reference gene validation is sound and consistent. To this end, using our qPCR workflow we have shown using two separate and unique datasets that conventionally chosen reference genes can render the same results of differential expression when compared to stable references selected from RNA-Seq data.

The advantage of this approach in filtering the best candidates from any conventional set lies in combining 3 methods that are complementary to each other. Although multiple statistical methods have been proposed before, we have previously highlighted the potential pitfalls and assumptions of these computational methods [12]. We thus devised this workflow combining the well-known NormFinder method with CV analysis and the visual representation of non-normalized expression followed by statistical testing. Indeed, it should be noted the core method of this approach is the NormFinder algorithm. However, this method was first described and validated with larger sample sizes (typically between 10–20 or more) and with a high number of candidate genes chosen from microarrays (typically between 15–20) exhibiting very little overall variation [8]. In our previous study, we identified that lower sample sizes and candidate genes with high overall variation can skew the results of this method because of the algorithm's construct [12]. Although scores of studies have used this algorithm without due diligence to the prerequisites, our combined approach resolves these issues. The CV analysis helps in the identification of genes with high overall variance and their successive elimination from analysis gives rise to more robust results from NormFinder. Visual representation of the

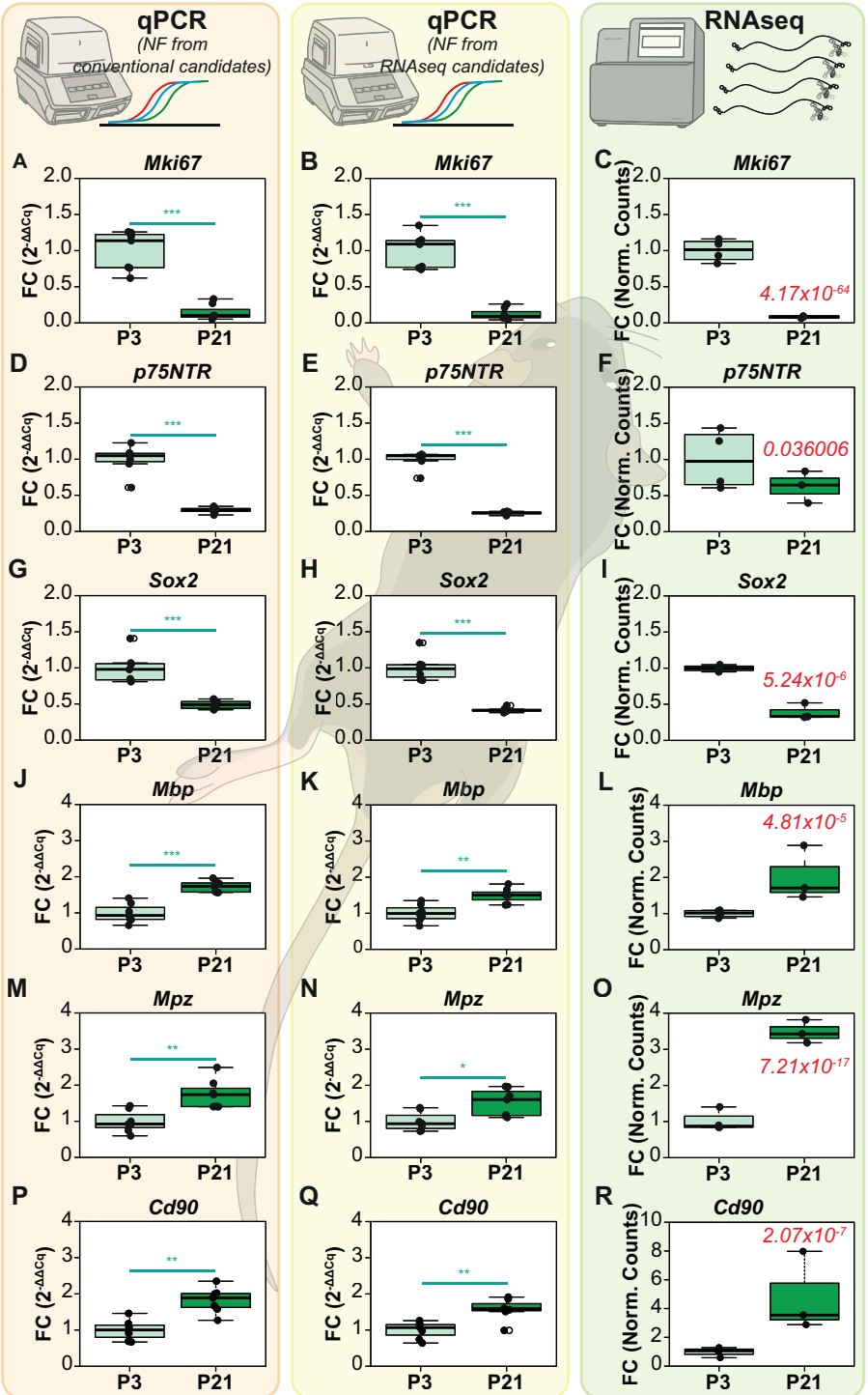

**Fig 6. Differential expression of target genes between P3 and P21 mouse sciatic nerves assessed by qPCRs and RNA-Seq.** The target genes (Mki67, p75NTR, Sox2, Mbp, Mpz, Cd90) have been normalized with the normalization factor (NF) computed from conventional reference genes Hsp60 & Ppia in the first column (A, D, G, J, M, P) or from RNA-Seq derived reference genes Ppp3ca and Coq9 in the second column (B, E, H, K, N, Q). The third column contains the Fold Changes computed from RNA-Seq (C,F,I,L,O,R). The normalized counts for each gene in each group were used to calculate the fold changes in RNA-Seq. The WT group was used as the experimental control. The $P_{adj}$ value of RNA-Seq is indicated in the P21 group for each gene in the RNA-Seq column. For the qPCRs, Non-parametric Mann Whitney U test was used to assess differences between the groups. The alpha value was set at 0.05 and P values are annotated as follows: * $P < 0.05$, ** $P < 0.01$, *** $P < 0.001$.

non-normalized profiles ($2^{-\Delta Cq}$) followed by statistical testing further validates the results of NormFinder as the Normalization Factor computed from the two best reference genes does not exhibit any significant differences between the experimental groups regardless of experimental conditions (**Figs 1K, 2K, 4K and 5K**). Thus, in combining CV analysis and statistical testing of non-normalized profiles, we extrapolated the NormFinder approach to be used for experimental setups involving lower sample sizes similar to those used in this study.

The cut off at 50% CV to qualify for NormFinder analysis is rooted in the fact that at 50%, the standard deviation of the expression distribution is half the mean expression level. Previous datasets that we published have indicated that the Normalization Factor (NF) computed by NormFinder when including genes that exhibit CV>50% exhibit significant intergroup variation in the non-normalized profiles [12]. Below this cut off, NF expression levels are stable across experimental groups. Indeed, the user can be more stringent by using a lower CV cut off but our data from this study and the previous one suggests that this is not necessary. However, this criterion gives rise to a theoretical limitation of our approach. In a given experimental setup, if all the candidate reference genes tested exhibit CV values of above 50%, then our approach will fail and NormFinder results would no longer be reliable. In such cases, a new list of candidates is required. However, our experience suggests that as little as 6 genes after eliminating genes exhibiting CV>50% can still provide a sound NF for qPCR data normalization [12].

Although not the primary objective of the study, the comparison of qPCR results with RNA-Seq merits discussion. In the iPSC dataset, comparisons made between qPCR and RNA-Seq results revealed that 4 out of 6 genes (*ACSS3*, *SYDE1*, *TREM2* and *TBX10*) showed no significant difference between the fold changes computed by both the methods (**S1D, S1E, S1F and S1C Fig**). However, changes in the expression of *CD52* and *P2RY12* significantly differed between the two methods (**Figs 3, S1A and S1B**). In the sciatic nerve dataset, 5 out of 6 genes tested show no difference between qPCR and RNA-Seq data in multiple comparisons (**S2A, S2B, S2C, S2E and S2F Fig**). However, this analysis is not optimal as the number of samples used in qPCR is much higher (N = 7) than the number of samples in RNA-Seq data that we mined (N = 3). In reality, only 3 genes (*Mki67*, *Sox2 & Mbp*) exhibited overlapping fold change distributions across the methods (**S2A, S2B and S2E Fig**). However, regardless of the magnitudes of the change observed, the pattern of expression changes was always conserved across qPCR and RNA-Seq for the genes that we tested. Indeed, it is possible that if we include more target genes, we can potentially notice disagreements in expression patterns as well. However, as stated earlier, our objective is restricted to demonstrating the futility of selecting reference genes from RNA-Seq data for qPCR data normalization. The comparison of the two methodologies falls beyond the scope of this study but it has been amply addressed elsewhere [23,38,39].

These studies have indeed tried to validate RNA-Seq data with RTqPCR for a large set of target genes typically ranging from a few hundred to the entire transcriptome. The consensus from these studies is that the extent of correlation between RTqPCR and RNAseq data is dependent on how the sequencing data was aligned, mapped and counted based on existing protocols. Additionally, though not discussed in these articles, we believe that this would equally depend on how reference genes were selected for qPCR assays that validate the RNA-seq data. Overlooking these technicalities, these studies also show that around 85% of all genes tested show concordant differential expression (comparable magnitudes with the same tendency) between the two methodologies with correlation coefficients typically above 0.8. Although these numbers are relatively reassuring at the outset, from an absolute perspective, the implications raise important concerns. For a hypothetical bulk RNA-Seq dataset of around 18000 protein-coding transcripts, 85% concordance leaves 2700 genes whose differential expression is discordant between RNA-Seq and qPCR. Of note, the study conducted by

Everaert and colleagues [23] conclude that discordant genes typically are smaller, have fewer exons and are expressed at lower levels. Indeed, and as mentioned in the introduction, these observations can be possibly substantiated by the transcript-length bias and the bias against genes with lower expression in RNA-Seq. In our study, the discordant genes *CD52* and *P2RY12* in the microglia dataset have only 2 exons. The other 4 concordant genes have between 4–13 exons and therefore are relatively larger. The basemean values (and therefore the expression levels) of the 2 discordant genes are largely higher than the 4 concordant genes. Therefore, number of exons appears to be more important than expression levels for concordance between qPCR and RNA-Seq for these target genes. On the contrary, we did not observe any correlation between discordance and small gene size or lower expression in the sciatic nerve samples. These observations show that an objective approach to identify discordant genes is lacking in literature. As RNA-Seq data analysis is particularly prone to certain biases while computing differential gene expression, it would be valuable to computationally suggest qPCR validation for specific genes that are susceptible to these normalization biases based on their transcript length, expression levels and other unidentified parameters. Ideally, these suggestions should be integrated into the data analysis pipelines and represented in the differential gene expression data frames. This would provide the foundation for an objective and structured validation of Sequencing data rather than an obtuse overall validation of RNA-Seq results by qPCR which is indeed burdensome. If these computational RNA-Seq validation suggestions are achieved, the methodology used in our study would aptly address the necessity of performing robust and reproducible qPCR assays using any conventional set of reference genes.

## Supporting information

**S1 Fig. Comparison of fold changes in the TREM2KO iPSC group among the three methods.** qPCR FCs were computed using Normalization factors from conventional candidates (NF Conv) or RNA-Seq derived candidate (NF RNA-Seq). RNA-Seq fold changes were calculated from the normalized counts. (A) P2RY12 (B) CD52 (C) TREM2 (D) ACSS3 (E) SYDE1 (F) TBX10. Non-parametric ANOVA was performed using the Kruskal Wallis test (red line comparing the means of the three groups). Multiple comparisons were performed using the Dunn's post hoc test using the NF conv. group as the control condition (green lines). The alpha value was set at 0.05 and P values are annotated as follows: * $P < 0.05$, ** $P < 0.01$, *** $P < 0.001$.
(EPS)

**S2 Fig. Comparison of fold changes in the P21 sciatic nerve group among the three methods.** qPCR FCs were computed using Normalization factors from conventional candidates (NF Conv) or RNA-Seq derived candidate (NF RNA-Seq). RNA-Seq fold changes were calculated from the normalized counts. (A) Mki67 (B) Sox2 (C) p75NTR (D) Cd90 (E) Mbp (F) Mpz. Non-parametric ANOVA was performed using the Kruskal Wallis test (red line comparing the means of the three groups). Multiple comparisons were performed using the Dunn's post hoc test using the NF conv. group as the control condition (green lines). The alpha value was set at 0.05 and P values are annotated as follows: * $P < 0.05$, ** $P < 0.01$, *** $P < 0.001$.
(EPS)

**S1 Table. RNA-Seq data features of the reference genes shortlisted from the sequencing data of WT vs TREM2KO iPSC microglia.**
(XLSX)

**S2 Table. RNA-Seq data features of the reference genes shortlisted from the sequencing data of P3 vs P21 mouse sciatic nerves.**
(XLSX)

## Acknowledgments

The authors thank the Animal House Core Facility and the Cyto2BM Molecular Biology Platform of BioMedTech Facilities (INSERM US36/CNRS UMS2009) for the animals and research services pertaining to the generation of qPCR data. We also thank Benjamin SAINTPIERRE and Dr. Franck LETOURNEUR from the Genomic Platform at Institut Cochin, Paris for the quality control of RNA samples and for their assistance in the analysis of RNAseq data. NKS would like to thank Prof. Christopher E SHAW and Tanisha LEWIS from the UK Dementia Research Institute, King's College London, for their support towards this work.

## Author Contributions

**Conceptualization:** Nirmal Kumar Sampathkumar, Venkat Krishnan Sundaram.

**Data curation:** Prakroothi S. Danthi.

**Formal analysis:** Nirmal Kumar Sampathkumar, Venkat Krishnan Sundaram, Prakroothi S. Danthi.

**Investigation:** Nirmal Kumar Sampathkumar, Venkat Krishnan Sundaram, Rasha Barakat, Shiden Solomon, Mrityunjoy Mondal, Ivo Carre, Tatiana El Jalkh, Aïda Padilla-Ferrer.

**Methodology:** Nirmal Kumar Sampathkumar, Venkat Krishnan Sundaram.

**Project administration:** Nirmal Kumar Sampathkumar, Venkat Krishnan Sundaram, Charbel Massaad, Jacqueline C. Mitchell.

**Validation:** Charbel Massaad, Jacqueline C. Mitchell.

**Visualization:** Mrityunjoy Mondal, Julien Grenier.

**Writing – original draft:** Nirmal Kumar Sampathkumar, Venkat Krishnan Sundaram.

**Writing – review & editing:** Nirmal Kumar Sampathkumar, Venkat Krishnan Sundaram, Prakroothi S. Danthi, Rasha Barakat, Shiden Solomon, Mrityunjoy Mondal, Ivo Carre, Tatiana El Jalkh, Aïda Padilla-Ferrer, Julien Grenier, Charbel Massaad, Jacqueline C. Mitchell.

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
