## [Decision Letter · Decision Letter 0]

26 Jan 2022

Dear Dr Sampathkumar,

We are pleased to inform you that your manuscript 'RNA-Seq is not required to determine stable reference genes for qPCR normalization' has been provisionally accepted for publication in PLOS Computational Biology.

Best regards,

Bjoern Peters

Benchmarking Editor

PLOS Computational Biology

Bjoern Peters

Benchmarking Editor

PLOS Computational Biology

Reviewer's Responses to Questions

**Comments to the Authors:**

Reviewer #1: All required changes to the manuscript have been performed.

Reviewer #2: The manuscript:"RNA-Seq is not required to determine stable reference genes for qPCR normalization" describe relevance of reference gene selection and validation. Even though RNA-Seq is suggested as potential solution for good reference identification, it is not working perfectly. Authors discuss selection drawbacks and show important results for broad community using gene expression analysis.

My previous comments (Review Commons) were addressed and also other comments from additional reviewers were answered. Already my previous review was positive and appreciated clear and logic text of manuscript and the main message. I recommend this manuscript for publication in Plos Computational Biology.

**Have the authors made all data and (if applicable) computational code underlying the findings in their manuscript fully available?**

Reviewer #1: Yes

Reviewer #2: Yes

PLOS authors have the option to publish the peer review history of their article (what does this mean?). If published, this will include your full peer review and any attached files.

Reviewer #1: No

Reviewer #2: **Yes: **Radek Sindelka

---

## [Editor Report · Acceptance letter]

22 Feb 2022

PCOMPBIOL-D-21-02155 

RNA-Seq is not required to determine stable reference genes for qPCR normalization

Dear Dr Sampathkumar,

I am pleased to inform you that your manuscript has been formally accepted for publication in PLOS Computational Biology. Your manuscript is now with our production department and you will be notified of the publication date in due course.

With kind regards,

Anita Estes
